# Can Prior Sexual Abuse Explain Global Differences in Measured Self-Esteem in Male and Female Adolescents?

**DOI:** 10.3390/children10020276

**Published:** 2023-01-31

**Authors:** Alice Sawyerr, Chris Adam-Bagley

**Affiliations:** 1Department for Continuing Education, University of Oxford, Oxford OX1 2JA, UK; 2Faculty of Social Sciences, University of Southampton, Southampton SO17 1UN, UK

**Keywords:** child sexual abuse, self-esteem, Rosenberg RSES, longitudinal studies, gender differences

## Abstract

World research has shown that adolescent and young adult women and girls have significantly “poorer” self-esteem than men and boys, on a variety of previously validated self-esteem measures. There is no consensus on reasons for this, and a variety of factors have been proposed: some adolescent girls have a preoccupation with facial and body features, and this leads to a global negative evaluation of self-characteristics; the measures themselves are biased towards describing self-characteristics on which men and boys are likely to evaluate themselves more favourably; and in an often-sexist world, women and girls experience (or anticipate) many structural disadvantages, in education, career and promotion, which lead girls to “internalise” an image of themselves as less able or worthy than men and boys. A separate literature on the sexual abuse and exploitation of children and adolescents has found that (a) sexual exploitation and maltreatment often has a sequel in impaired self-concept and self-esteem and (b) sexual maltreatment is twice as likely to occur in women and girls. It is puzzling that differential levels of child sexual abuse have not been advanced in many studies as an explanation of gender differences in self-esteem in the large-scale studies we review, although this effect is confirmed by clinical and social work literature.

## 1. Introduction: Concepts of the Self, and Measurement of Self-Esteem

The concept of “the self” in modern psychology has been developed from the ideas of American educationist William James:

*The world experienced (otherwise called “the field of consciousness”) comes at all times with our body as its center of vision, center of action, center of interest. Where the body is, is “here”; when the body acts is “now”; what the body touches is “this”; all other things are “there” and “then” and “that”. These words of emphasized position imply a systematization of things with reference to a focus of action and interest that lies in the body…. The body is the storm center, the origin of coordinates, the constant place of stress in all that experience-train. Everything circles round it, and is felt from its point of view. The word “I” then, is primarily a noun of position, just like “this” and “there”*.(William James, 1890, pp. 154–155) [1].

Woźniak [2] shows how the work of James influenced the development of humanistic psychology, and the “I”:“me” dichotomy, which has particular importance in how sexually abused children and adults develop an adult identity [3,4]. In drawing the strands of identity theory together, we have argued that *self-concept* defines those aspects of oneself that are regarded as relevant and important, including not only gender and ethnicity but also the whole nexus of roles which one is required to perform. *Self-esteem* describes how one evaluates performance in these roles. *Identity* is the sum total, at any point in time, of personal roles together with how one reflects the evaluation of role performance, by oneself and others [5,6]. Reference groups in acquiring self-esteem are important, as Rosenberg and Simmons showed in their comparison of self-esteem in African-American and European-American youth. Contrary to expectation, African-American youth had equal or better self-esteem than other ethnic groups because the sources of their self-evaluation were neighbourhood and family, and various role models in the US’s Black community [7]. However, in this same work, Rosenberg and his colleagues show that women and girls in all ethnic groups had “poorer” self-esteem than did men and boys, a phenomenon that has yet to be satisfactorily explained.

We stress, in the quotation from William James above, the importance of “me” which is embedded in a young person’s physical body. If the child’s sexual body has been violated or prematurely stimulated, then the balance between “I” and “me” which characterises normal child development has also been violated or disrupted [3]. Serious other role violations include breaking the incest taboo, which can have catastrophic consequences for the immature partner (who is almost always female) [8], and the violation of a young boy’s heterosexual identity when he is sexually exploited by an older male [9].

The measurement of global self-esteem began to be systematised some 50 years ago, beginning with work by Piers-Harris and Coopersmith [10,11], and most recently and definitively in the development and validation of the short (10-item) Rosenberg scale [12]. These various self-esteem scales correlate with one another at about 0.6 or greater and appear to be measuring the same construct [13]. Negative self-esteem belongs to the same psychological dimension as anxiety and depression. Poor self-esteem often indicates a depressed and querulous appraisal of self-characteristics and a lack of confidence in role performance. The 10-item Rosenberg Self-Esteem Scale (RSES) is now the most widely used scale across the world, has been translated into 28 languages and is a reliable and valid instrument (in terms of factor structure and face validity) in many cultures [14]. The RSES has also been validated through its correlation with mental health measures in large samples of high school students in Canada, England and Hong Kong [15,16,17]. The test–retest reliability of the scale is excellent over the lifespan [12,18]: on average, scale scores decline with ageing.

However, very high scorers whose RSES scores do not decline with age indicate a personality type called “grandiose” or “narcissist”, as opposed to the reflective quality of successful ageing [19,20]. The opposite “syndrome” is that of “devastated self-esteem”, in which chronically depressed individuals have lifelong patterns of severe depression and very low self-esteem, which are enduring across the lifespan [21,22]. Dysfunctional interactions of various kinds were mentioned in a 2001 review of self-esteem theory and research, as potential predictors of low self-esteem in adulthood [23], but the possible impacts of sexual abuse per se was not mentioned, despite the growing body of research linking child sexual abuse (CSA) with impaired self-esteem and to various psychiatric conditions in adulthood.

### 1.1. Child Sexual Abuse and Its Impact on Impaired Self-Esteem, and Mental Health Adjustment: Earlier English and Canadian Studies

Western cultures and social scientists have recognised child sexual abuse (CSA) as a major problem only in the past 50 years. The issue had been addressed in the clinical literature, but a review of incestuous abuse published in 1969 in the American journal *Social Problems* could find no published epidemiological studies of mental health outcomes [8]. This review prompted pioneering studies in the UK and Canada, which showed that surveys of clinical groups suggested that early childhood exploitation and abuse was a frequent cause of poor mental health in adulthood [24,25]. A different kind of study, surveying random samples of young adult populations which asked adults to recall events of sexual abuse in childhood and adolescence established both prevalence figures and established mental health profiles in these nonclinical populations. About 10% of women and girls, and about 5% of men and boys recalled “serious” sexual abuse up to their 16th birthdays (unwanted events of handling of intimate parts of the body and/or anal or vaginal penetration) [26]. The longer the abuse occurred and/or the number of abusive episodes, the poorer were the psychological outcomes for victims [27,28,29]. Sexual abuse, combined with within-family emotional abuse, and physical abuse (from whatever sources) often had a synergistic effect, so the worst psychological outcomes were for long-term abuse victims of the most serious forms of abuse, combined with emotional and/or physical abuse [30,31,32]. Clinical outcomes in this first phase of mapping the extent of and damage from child sexual abuse were measured mainly in terms of depression, anxiety disorders, and suicidality. Low self-esteem levels (in early Canadian work, which included brief measures of prior abuse at home, in the neighbourhood and at school) were associated with higher scores on measures of various forms of mental ill-health and poorer self-esteem [33,34,35]. For women and girls in these earlier studies, the large majority of the abusers were men (98%), with the same proportion for assaults on men and boys [29]. These were not “homosexual” assaults per se but rather assaults by adult male paedophiles who sometimes exploited young adolescent boys who were experimenting with “coming out” into gay roles [36].

These epidemiological studies of geographically stable populations in Canada were likely to be underestimates given that some female victims who had run from abusive homes were likely to be pulled into the adolescent sex trade industry, in which abuse was piled on abuse [37,38].

### 1.2. Recent Psychiatric, Psychological and Social Work Studies in Child Sexual Abuse and Polyvictimisation

Despite the failure of “mainstream” social psychology to take into account the incidence and impact of sexual abuse in childhood and adolescents on individual self-esteem, there is growing literature, with sound methodologies, which does make such a link. This literature (published between 2003 and 2015) supports and elucidates earlier findings and observations from American, English and Canadian research:


*A narrative review of the findings of world research on the sexual abuse of children, and its behavioural and mental health sequels in adolescent and adult adjustment (abuse which frequently occurs within family settings) … shows that around 9% of all women, and about 3% of all men have experienced prolonged, bodily intrusive abuse by the age of 16 or 18. This has many adverse sequels including impaired self-esteem, clinical levels of depression and anxiety, self-harm and substance abuse, somatic disorders, and many forms of maladaptation. Poly-victimisation combining physical, sexual and emotional abuse has particularly negative impacts. The long-term burden in human suffering and public health costs is high. In school, abuse victims are often bullied and isolated, which exacerbates (or even triggers) the negative effects of abuse. Teachers and school counsellors and social workers have an important role to play in identifying abuse victims, and offering help in ways which prevents the development of serious mental health problems.*
[39]

In this reviewed literature, an important study stands out, so we give further details of it here: Bellis and colleagues [40,41,42] used a public health model in designing and analysing the results of a national English survey of 3885 adults aged 18 to 69, focusing on the recall of “adverse childhood events” (ACES): 8.3% of those surveyed had experienced four or more adverse events in childhood. Such a history was associated with a greatly elevated risk of poor adult mental health, substance abuse and behavioural problems, including criminality in the adult lives of those who had experienced highly stressed childhoods. These problems were also particularly likely to occur in those from deprived urban areas. Using a public health model, Bellis and colleagues [40,41,42] found that ACES accounted for 12% of adult alcoholism; 5% of adult hard drug use; and 38% of unwanted teen pregnancy. Concerning mental health sequels, *“Links between such behaviours and childhood circumstances are likely to operate through the impact of ACES on the developing brain and its stress regulatory systems, which affect factors such as emotional regulation and fear response, and this may predispose individuals to health-harming behaviours”* (p. 90) [41].

Because our overview of child sexual abuse and abuse within the broader context of ACES was published in 2017, further research on the incidence and correlates of sexual and other abuse inflicted on children supports earlier clinical profiles showing that adult survivors of early sexual abuse and polyvictimisation have particularly poor mental health outcomes [27,43]. However, the outcome measures in many of the reviewed studies were of levels of depression, anxiety and somatic disorders, not self-esteem per se. However, because previous clinical work has indicated a strong correlation between impaired self-esteem and the disorders of depression and anxiety [39], it is reasonable to assume that the abused individuals in these large-scale, epidemiological studies would have poorer self-esteem.

Published research suggests that many children in disorganised families in which a woman’s male partners “come and go” are at elevated risk of sexual exploitation by both father figures and older siblings. These vulnerable children are also at risk of emotional and physical abuse both within and outside of the family dwelling (or “care home”). In this research on a UK longitudinal study of children followed up to young adulthood, parents gave information on their child’s experience of physical, emotional and sexual harm. Parents in an English longitudinal study often admitted to their imperfections as carers in terms of physical and emotional care, and they identified polyvictimisation (the overlap of physical, emotional and sexual abuse) [43]. Commenting on these English findings, Finkelhor and Tucker (2015) observe that studies in many cultures indicate that *“Molestation, rape, exposure to domestic violence, corporal punishment, physical abuse, bullying, sexual molestation … all overlap. ‘Peer violence’ is part of this ‘maltreatment syndrome’”* (p. 481) [44]. It is important to observe that David Finkelhor, the leading US scholar researching child sexual abuse, has been writing and publishing in leading journals and monographs about the nature and impact of child sexual abuse and its psychological correlates in the short and long term since 1979 [45,46,47]. A question must be asked: why have other researchers using Rosenberg’s RSES ignored and continued to neglect (until the second decade of the present century) the importance of prior, prolonged child abuse as a factor leading, in at least half of victims, to permanently impaired self-esteem?

Studies of another large UK birth cohort published by public health researchers Bentivegna and colleagues in 2022 indicated the following:

*“Sexual violence during mid-adolescence contributes markedly to the high prevalence of mental ill health observed in adolescents, especially in girls, and actively tackling sexual violence could lead to substantial improvements in adolescent mental health. In a UK report published in 2021, estimates from school populations showed high levels of sexual violence, with 79% of the surveyed girls reporting sexual assault of any kind. Sexual violence during mid-adolescence contributes markedly to the high prevalence of mental ill health observed in adolescents, especially in girls, and actively tackling sexual violence could lead to substantial improvements in adolescent mental health”*.[32]

These cohort and epidemiological studies have not usually focused on impaired self-esteem as an outcome of experiencing sexual abuse and other forms of child maltreatment, although it is reasonable to infer from the previous literature on correlations of self-esteem with clinical profiles of depression and anxiety that the individuals studied would have impaired self-esteem levels. In a Canadian longitudinal study [30,31] it was shown that not only was this the case but often when impaired self-esteem occurred in disorganised families, socialisation problems (including emotional abuse and excessive physical punishment) often *preceded* a child’s sexual exploitation. There was effectively a synergistic effect in which vulnerable children with low self-esteem experienced sexual victimisation and sexist bullying at home and in school, with outcomes including very low self-esteem and associated high levels of depression, suicidality and other psychological disorders.

Okunlola and colleagues (2021) [48] reviewed world literature published from 2007 to 2018 and identified eight representative studies in different world cultures which measured self-esteem (most using the RSES). In six of these eight studies, sexual abuse history was significantly linked to impaired self-esteem, and the effect was very clear for the most serious forms of abuse. Significant intervening variables were gender (greater impact in women and girls), lower social status and family disruption. Thus, self-esteem outcomes following sexual abuse were greatest in girls from disorganised families, in which sexual exploitation was linked to both stigmatisation and self-blame. In a Canadian study in 2021 [49] on prior sexual abuse that reported on a cohort of 8194, aged 14–18, prior sexual abuse, while strongly impacting self-esteem levels in girls, did also negatively affect self-esteem in boys, who often reacted to CSA in entering delinquent pathways.

Most of the studies reviewed come from the US and Canada, whose scholars continue to be the main contributors to this field. For example, Reid-Russell and her colleagues (2021) [50] followed up 240 8- to 16-year-olds for 2 years: half had been physically, emotionally or sexually abused. A measure of “implicit self-esteem” identified abuse victims, and these victims had much higher levels of depression and suicidal ideation. Measurements taken 2 years apart showed that abuse had an early impact on self-esteem, and it was low self-esteem levels which were linked to subsequent depression and suicidal feelings. Boys and girls were not separated in data analysis, so this effect may apply to both genders.

It is established, then, that there is a reliable corpus of knowledge showing that sexual abuse and other types of abuse of children and adolescents occur in all world cultures, and experiencing these assaults is linked to depressed levels of self-esteem (which in turn leads to vulnerability to other stressors, with resulting depression and other mental health problems). These findings apply to both genders. Explanations of this link require “gendered pathways” in longitudinal research [50]. It seems realistic to propose that measured self-esteem differences between genders is due, in part or whole, both to the extent of prior abuse, including polyvictimisation; to disorganised family living, resulting in ACES histories; and to the relationship of the abuser to the victim.

We concluded a review of studies on child sexual abuse, including our own, available up to 2016:

*While the British researchers, who are mainly child and adult psychiatrists, focus on diagnosable mental health conditions, there are grounds for assuming that an individual with depression, anxiety and self-harm falling into the clinical domain, would also have very low self-esteem. In terms of William James’ (1890) model of the body being at the centre of self-concept and self-esteem: the violation of the child’s body through physical and sexual abuse interferes profoundly with core developments of the self-concept, and the evaluative construct of self-esteem, the innate feelings of “goodness” or “badness” by which the child develops a self-schema. The abused child acquires an identity which moves on a particular trajectory, one of watchfulness, fear, nightmare, nervousness, a pattern “burned into the brain”, even causing permanent structural brain changes which last a lifetime*.[39]

The issue of whether the fact that higher rates of sexual abuse experienced by girls can account (in whole or part) for the different levels of self-esteem observed between genders remains an open question for mainstream psychologists.

### 1.3. “Mainstream” Social Psychology and Explanations of Gender Differences in Self-Esteem

A review of US theory and research on the nature and origins of self-esteem was gathered in a book edited by Rosenberg and Kaplan in 1982 [51]. This book contained 42 chapters summarising the insights generated by “mainstream” US research up to 1981. Although a literature describing the nature and extent of child sexual abuse, its negative impact on mental health and its potential effects in depressing self-esteem in the US was emerging before that date [46,47], *not a single chapter* in this book mentions child abuse of any kind as a possible correlate of impaired self-esteem. This is puzzling. Three chapters of the book [52,53,54] identify gender as a factor in impaired self-esteem, but none of them mentions child maltreatment or abuse as a potential cause. The chapter by Rosenberg [52] reports that by age 15, girls’ self-esteem (on the RSES scale) in large US samples was about 7% lower than that of boys, a difference attributed to enhanced levels of self-consciousness and self-criticism in women and girls. Kagan attributes lower self-esteem in girls to their ascribed “sex role identity”, in that girls tend to internalise the prevailing sexist stereotypes about female roles [53]. Cox and Bauer [54] report experimental data showing that adult women are easier to control and manipulate, and women’s greater desire for social approval and their acceptance of society’s stereotypes feed into a more negative self-concept.

Nowhere in this edited book is there any mention of child maltreatment, even though evidence on such issues and their negative impact on mental health and self-esteem was emerging in the US and Canada by the late 1970s. However, these latter studies were addressed to the medical, counselling and social work professions, and they failed to gain the attention of mainstream academic social psychologists such as Morris Rosenberg and his peers.

A sequel to this edited volume on studies elaborating work using the Rosenberg and other self-esteem scales was published in the US in 2001, edited by Owens et al., titled *Extending Self-Esteem Theory and Research: Sociological and Psychological Currents* [55]. The aim of this book’s 18 chapters was to consolidate the evidence on Rosenberg’s RSES and to point the way forward to new research. By this time, there was now abundant evidence from US and Canadian studies on the role of child abuse and child maltreatment in depressing self-esteem, with subsequent mental health problems—the authoritative work of John Briere, in monographs and journal articles published in the 1990s [56,57] had now been added to David Finkelhor’s studies [45,46,47]. However, *not a single chapter* in the 2001 edited volume on self-esteem research [55] mentions child abuse of any kind as a factor that might negatively impact self-esteem. Chapters in this book on family factors affecting self-esteem levels in children, adolescents and young women did not consider child abuse or maltreatment as a variable [58,59], and a chapter on deviant youth failed to consider maltreatment as a factor influencing self-esteem in boys and their subsequent “deviant behaviour” [60]. A chapter by Rosenberg and Owens [61] reported on a longitudinal study from childhood to midlife in which low self-esteem children become chronically depressed and marginal individuals in society, with high levels of expressed suicidality. The authors point to negative interactions between self-schema and reaction to stresses in these individuals, but causes in childhood trauma were not considered.

Even after child sexual abuse was established, by the year 2000, as a major factor in depressing self-esteem in young girls and was likely to be enduring into adulthood [45,46,47,56,57,62], most US studies of self-esteem and its enhancement in girls continued to ignore the reality and impact of child sexual abuse. For example, a leading popular publication in the US (*Psychology Today*) in a 2022 article titled “Low self-esteem in adolescents: what are the causes?” [63] lists several potential causes, including the trauma of being in or witnessing an auto accident and witnessing violence of various kinds. However, the article *entirely fails* to address issues of sexual assault (and other forms of maltreatment) experienced by children and adolescents.

A group of 30 female psychologists who were members of the American Psychological Association (APA, an important body in the registration and professional education of psychologists) contributed to a publication in 2008 on *A New Look at Adolescent Girls* [64]. These psychologists commented on a variety of issues and challenges facing female adolescents in the US, including body image, negative messages from electronic media and factors discouraging women from taking leadership roles in industry, management and professions. A section on sexual issues and concerns for adolescent women *failed* to address the fact that up to 10% of girls have been subjected to prolonged sexual abuse, which often has a strong negative impact on adjustment and self-esteem. Another publication on “gender gaps in self-esteem” in 2016 by the American Psychological Association [65] commented on cultural factors which might explain gender differences in self-esteem, but it failed to make any mention of prior sexual abuse to explain these differences. So far as we can locate, it was not until 2020 that the APA offered professional guidelines for coping with child maltreatment (including sexual abuse) and its aftermath, which included “diminished self-worth”: in this year, the APA issued a balanced practice guide to the now-impossible-to-ignore clinical literature on the long-term effects of sexual and other kinds of abuse imposed on children and adolescents [66]. In 2021, the APA issued a professional briefing on “child abuse and neglect” and included sexual abuse as one form of maltreatment [67].

The failure to include abuse and maltreatment in large-scale studies on self-esteem in children, adolescents and young adults in the US was *not* because there had been a dearth of research studies in lower self-esteem levels in women and girls. A frequently cited review of research by Kling et al. (1999) [68] examined 216 US studies which reported consistent differences in self-esteem in women and girls, larger differences being recorded in 14- to 18-year-olds. They surmised, without supporting data, that young women put greater stress than men on presenting themselves as physically attractive, their subjective failure to achieve this being reflected in poorer self-esteem. Studies showing the impact of child abuse on self-esteem, and the differential incidence and effects of such abuse according to gender, were not discussed in this review, an omission that remained in mainstream US professional psychology in general and in social psychology in particular for almost 2 decades.

A further meta-analysis of US studies on self-esteem was published by Gentle et al. in 2009 [69], examining 428 studies, which included 32,486 young Americans who completed various measures of self-esteem (Piers-Harris, Coopersmith, Rosenberg and Tennessee scales). Overall, men and boys had “better” self-esteem than women and girls, but in different domains, including those relating to athletic prowess and achievement of success in traditional male domains. Women appraised themselves more favourably in moral-ethical domains, areas relating to “good” behaviour. The authors concluded that poorer overall levels of self-esteem in women and girls were due to “reflected appraisals”, of how young American women were expected to behave. Issues of child abuse were not addressed in this review, in that they were not mentioned in any of the studies that these authors reviewed.

Further US studies on self-esteem failed, again, to consider the potential impact of child abuse on long-term self-esteem development in men and boys and in women and girls. Bachman et al. in 2011 [70], for example, reported findings for the Rosenberg RSES scale for American adolescents aged 14 (N, 102,109), 16 (N, 107,849) and 18 (N, 107,421) on the basis of high school sample data collected between 1991 and 2008. The RSES means were similar in each year of study. In all ethnic groups, women and girls had significantly poorer self-esteem than men and boys, but the authors did not advance any particular theory to account for this gender difference.

Erol and Orth (2011) [18] reviewed all available studies in order to identify factors “known to be associated with self-esteem levels”, in large samples from a variety of cultures. These included gender (in 21 studies), where women and girls had significantly lower mean scores (i.e., poorer self-esteem) than men and boys had. Erol and Orth [18] gave results from their own study of 7100 Americans systematically tracked from ages 14 to 30. At each age, RSES correlations in each ethnic and gender group were similar: “high self-esteem” individuals were more likely to be extroverted, emotionally stable, conscientious, low in risk taking and in better physical health. The strongest predictor of self-esteem was a sense of satisfaction in reaching one’s goals at any point in time. Differences in self-esteem between genders remained significant within all ethnic groups, after various psychological variables were controlled for. However, in this, as in the other large-scale US studies, *abusive or traumatic events in childhood were not measured*.

Bleidorn et al. (2015) [71] report the results from an international study on gender and self-esteem, which developed a large (but probably nonrandom) data set (N, 985,937 from 48 countries), based on an internet study asking “young people” to report on a variety of problems they were experiencing, without having to identify themselves. Respondents were also invited to anonymously complete a brief self-esteem scale (based on the RSES) and to give details on their cultural situation and circumstances and on their home country. Respondents were aged 16 to 45 (49% female). The self-esteem measure was expressed as a standardised score with a mean of 50. For all cultures and age groups, men and boys scored 1.85 points higher (at 51.85) than did women and girls—a highly significant difference. The authors offer a variety of speculations to explain these differences but do not mention child abuse or maltreatment, which they did not measure (although this could have been performed in a survey which guaranteed anonymity). They commented, however (without additional evidence), that women may have “lower” self-esteem levels because of prevailing sexist norms, which embody stereotypes that women and girls are socialised to adopt. It seems that self-esteem impairment because of child abuse, and the differential effects on women and girls who experience sexual abuse in childhood, has only atypically been part of the focus of mainstream psychology in the US, up to 2018 [72].

## 2. Conclusions

This review leads us to conclude that: (a)Child sexual abuse leads to long-term changes in mental health and self-identity, including impaired self-esteem;(b)The prevalence of prior sexual abuse in females is at least twice as great for females, than for males; (c)Studies have consistely shown that in large non-clinical populations, males have significantly ‘better’ self-esteem than do females: and the prevalence of prior sexual abuse in females is at least twice the prevalence in males; (d)It is a distinct but yet unmeasured possibility that lower self-esteem rates in females are almost entirely due to the impact of sexual abuse, a hypothesis that has largely been ignored by mainstream psychology (but not by humanist psychology).

Our hypothesis for future research, derived from this review, is that when the effect of prior sexual is controlled for, measured self-esteem differences between genders will *fail to be statistically significant*. While this hypothesis was tentatively supported in a large population of English high school students [6], the sampled population was overweighted by ethnic minorities with African-Caribbean or Asian heritage, so the results cannot be a definitive test of the hypothesis that self-esteem differences between genders will disappear when prior sexual abuse is controlled for. New studies are required in this important area of research.

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
