# Peer review of "Can Prior Sexual Abuse Explain Global Differences in Measured Self-Esteem in Male and Female Adolescents?"

_children, 2023, doi:10.3390/children10020276_

Round 1

Reviewer 1 Report

This study aimed to examine "global" gender differences in self-esteem by using 1,985 adolescent students from 10 schools in Northern England. Obviously, results obtained from this sample cannot be generalized to "global" samples, nor should you conclude that "a single variable [childhood sexual abuse] could explain all of the gender difference in self-esteem levels." Instead, your findings provide support for further research into your hypothesis that a history of CSA may explain gender-based differences in self-esteem. 

I have made extensive suggestions for revising and reducing the length of your introduction, and also for ways to analyze your data. I hope you find them helpful.   

Author Response

I would like to thank both reviewers for their helpful comments.

Response to comments of Reviewer 1

Line 38:               Suggested text corrections have been made.

Line 46:                Textual clarification requested: this has been addressed.

Lines 58-60:        Be more explicit about the purpose of the study: this has ben addressed, with expanded explanation.

Line 66:                “Sexual identity” added as a concept, as suggested.

Line 82:                A short section deleted here, with added clarification.

Line 85-6:            Textual clarifications made.  We have used English, rather than American forms of spelling.

Line 109:             Clarified, as suggested.

Lines 120-5:        Section excluded, as recommended.

Line 126:             Statement modified to be more tentative, as recommended.

Line 144:             We have tried to justify the focus on ACES (Adverse Childhood Experiences)

Line 147:             Out textual error has been corrected.

Line 177:             Text amended as suggested, to focus on “sexual abuse”

Line 182:             The valuable suggestion to move literature cited later into this part of the paper has been followed.

Line 202:             Our assumptions about cause and effect have been modified, as suggested.

Line 212:             These references moved from later in the paper to here, as suggested.

Line 226:             The statements have been modified so as to be more tentative, as suggested.

Line 242:             We have introduced the Foner reference to “shame” experienced by victims, as suggested.

Line 259:             Suggestion for moving literature on self-esteem and gender has been followed. This makes sense, and clarifies the review.

Line 280:             The reviewer asks why we are focussing on historical trends in literature on sexual abuse and self-esteem. Our answer is that right up to 2018 mainstream psychology  often ignored links between sexual abuse, gender and self-esteem: this is an important issue not only in the history of modern clinical and social psychology, but it also has implications for how sexual abuse is perceived and prevented, and its victims treated.

Lines 321-3:        We have accepted this suggestion, and moved part of this review into an earlier section.

Line 378:             Our revisions have shortened the paper, as suggested.

Line 403:             Yes, “absent” has been inserted.

Line 439:             We agree, and the text has been amended.

Line 455:             Correction accepted.

Line 457:             We think we have addressed the problem of which t-tests to interpret by applying ANCOVA (combing regression and variance analysis), covarying gender by self-esteem, while controlling for the covariation of sexual abuse history. (This valuable approach to data analysis was suggested by Reviewer 2). We stress more strongly that these are tentative results, which require replication.

Line 465:             Amended, as suggested.

Line 614:             References are being amended to conform to MDPI style requirements.

Note:                    I am unable to upload the original file containing Reviewer 1’s comments, since I can’t upload the PDF file containing them,  into a Word File. Since the paper has now been amended and shortened, the above lines are no longer relevant in the revised version.

Reviewer 2 Report

This manuscript focuses on an important topic and a notable gap in the literature on gender differences in adolescent self-esteem. Overall, the paper provides a lot of specific information in the introduction at the expense of more detailed information in the other sections of the paper. I think the introduction could be made more concise. I have listed some additional specific comments below.

Intro: Line 219 says “most had been abused sexually, emotionally or sexually” I think you mean to include physically in there as well.

The Methods section needs a data analysis section detailing the planned analyses.

Results: I was confused by the columns in the table because sexual abuse was delineated by frequency (CSA 1-2 times, CSA 3+ times). My impression from the methods section was that the CSA measure was a yes/no question. Please provide more detail about the assessment of CSA and the response choices participants were given.

Results: It looks like you did four separate t tests to examine your hypothesis of whether males and females significantly differed in levels of self-esteem while controlling for sexual abuse history. This strategy does not effectively answer your question. I think an ANCOVA would be more appropriate so you can analyze mean self-esteem differences across genders while statistically controlling for sexual abuse history in one test.

The organization of the paper is at times difficult to follow. For example you discuss the referrals for counseling given to study participants in your discussion. This seemed like an abrupt shift from the previous paragraph and this content belongs in the methods section. Another example: you provide details about the measures administered in two different sections of the methods. Please review the manuscript to ensure that content is in the relevant section defined by an appropriate header.

Author Response

I would like to thank both reviewers for their helpful comments.

This Reviewer offers general comments, rather than line-by-line observations.  In response:

Line 219:             This semantic error has been corrected.

A New Methods section has been identified, and largely irrelevant material and discussion (on counselling victims) has been deleted.

Results:               The Yes/No question on experience of sexual abuse -  this was the response to each event of abuse, and individuals sometimes experienced further abusive events, accounting for the “3+” designation.

Data analysis:     We accept that the use of ANCOVA is the way to clarify the task of interpreting the t-test data. We have undertaken this, and the resulting ANCOVA gives a clear result supporting the inferences from the t-test.

Round 2

Reviewer 1 Report

I appreciate the minor changes you made in the paper's organization. However, I do not believe it is improved enough to warrant publication. The major problem is the way you analyze the data and present your results. You either need to have a statistical consultant to advise you on analyzing your data, or else delete the data/study and instead, write a review of the peer-reviewed research.  I am attaching my comments on the manuscript for you to use to revise. 

Author Response

Thank you. Please find the revised version attached.

Round 3

Reviewer 1 Report

Good suggestions for future research